# Positive Curvature and Hamiltonian Monte Carlo

**Christof Seiler**    **Simon Rubinstein-Salzedo**$^{*}$    **Susan Holmes**
Department of Statistics
Stanford University
{cseiler,simonr}@stanford.edu, susan@stat.stanford.edu

## Abstract

The Jacobi metric introduced in mathematical physics can be used to analyze Hamiltonian Monte Carlo (HMC). In a geometrical setting, each step of HMC corresponds to a geodesic on a Riemannian manifold with a Jacobi metric. Our calculation of the sectional curvature of this HMC manifold allows us to see that it is positive in cases such as sampling from a high dimensional multivariate Gaussian. We show that positive curvature can be used to prove theoretical concentration results for HMC Markov chains.

## 1   Introduction

In many important applications, we are faced with the problem of sampling from high dimensional probability measures [19]. For example, in computational anatomy [8], the goal is to estimate deformations between patient anatomies observed from medical images (e.g. CT and MRI). These deformations are then analyzed for geometric differences between patient groups, for instance in cases where one group of patients has a certain disease, and the other group are healthy. The anatomical deformations of interest have very high effective dimensionality. Each voxel of the image has essentially three degrees of freedom, although prior knowledge about spatial smoothness helps regularize the estimation problem and narrow down the effective degrees of freedom. Recently, several authors formulated Bayesian approaches for this type of inverse problem [1, 2, 4], turning computational anatomy into a high dimensional sampling problem.

Most high dimensional sampling problems have intractable normalizing constants. Therefore to draw multiple samples we have to resort to general Markov chain Monte Carlo (MCMC) algorithms. Many such algorithms scale poorly with the number of dimensions. One exception is Hamiltonian Monte Carlo (HMC). For example, in computational anatomy, various authors [22, 23] have used HMC to sample anatomical deformations efficiently. Unfortunately, the theoretical aspects of HMC are largely unexplored, although some recent work addresses the important question of how to choose the numerical parameters in HMC optimally [3, 7].

### 1.1   Main Result

In this paper, we present a theoretical analysis of HMC. As a first step toward a full theoretical analysis of HMC in the context of computational anatomy [22, 23], we focus our attention on the numerical calculation of the expectation

$$I = \int_{\mathbb{R}^d} f(q)\,\pi(dq) \tag{1.1}$$

---

$^{*}$The first and second authors made equal contributions and should be considered co-first authors.

by drawing samples $(X_1, X_2, \dots)$ from $\pi$ using HMC, and then approximating the integral by the sample mean of the chain:

$$\widehat{I} = \frac{1}{T} \sum_{k=T_0+1}^{T_0+T} f(X_k). \tag{1.2}$$

Here, $T_0$ is the burn-in time, a certain number of steps taken in the chain that we discard due to the influence of the starting state, and $T$ is the running time, the number of steps in the chain that we need to take to obtain a representative sample of the actual measure. Our main result quantifies how large $T$ must be in order to obtain a good approximation to the above stated integral through its sample mean ($V^2$ will be defined in §3, and $\kappa$ in the next paragraph):

$$\mathbb{P}(|I - \widehat{I}| \geq r\|f\|_{\text{Lip}}) \leq 2e^{-r^2/(16V^2(\kappa,T))}.$$

The most interesting part of this result is the use of *coarse Ricci curvature* $\kappa$. Following on ideas from Sturm [20, 21], Ollivier introduced $\kappa$ to quantify the curvature of a Markov chain [16]. Joulin and Ollivier [12] used this concept of curvature to calculate new error bounds and concentration inequalities for a wide range of MCMC algorithms. Their work links MCMC to Riemannian geometry; this link is our main tool for analyzing HMC.

Our key idea is to recast the analysis of HMC as a problem in Riemannian geometry by using the Jacobi metric. In high dimensional settings, we are able to make simplifications that allow us to calculate distributions of curvatures on the Riemannian manifold associated to HMC. This distribution is then used to calculate $\kappa$ and thus concentration inequalities. Our results hold in high dimensions (large $d$) and for Markov chains with positive curvature.

The Jacobi metric connects seemingly different problems and enables us to transform a sampling problem into a geometrical problem. It has been known since Jacobi [10] that Hamiltonian flows correspond to geodesics on certain Riemannian manifolds. The Jacobi metric has been successfully used in the study of phase transitions in physics; for a book-length account see [17]. In probability and statistics, the Jacobi metric has been mentioned in the rejoinder of [7] as an area of research promise.

The Jacobi metric enables us to distort space according to a probability distribution. This idea is familiar to statisticians in the simple case of using the inverse cumulative distribution function to distort uniformly spaced points into points from another distribution. When we want to sample $y \in \mathbb{R}$ from a distribution with cumulative distribution function $F$ we can pick a uniform random number $x \in [0, 1]$ and let $y$ be the largest number so that $F(y) \leq x$. Here we are shrinking the regions of low density so that they are less likely to be selected.

## 1.2 Structure of the Paper

After introducing basic concepts from Riemannian geometry, we recast HMC into the Riemmanian setting, i.e. as geodesics on Riemannian manifolds (§2). This provides the necessary language to state and prove that HMC has positive sectional curvature in high dimensions, in certain settings. We then state the main concentration inequality from [12] (§3). Finally, we show how this concentration inequality can be applied to quantify running times of HMC for the multivariate Gaussian in 100 dimensions (§4).

## 2 Sectional Curvature of Hamiltonian Monte Carlo

### 2.1 Riemannian Manifolds

We now introduce some basic differential and Riemannian geometry that is useful in describing HMC; we will leave the more subtle points about curvature of manifolds and probability measures for §2.3. This apparatus will allow us to interpret solutions to Hamiltonian equations as geodesic flows on Riemannian manifolds. We sketch this approach out briefly here, avoiding generality and precision, but we invite the interested reader to consult [5] or a similar reference for a more thorough exposition.

*Definition* 2.1. Let $\mathcal{X}$ be a $d$-dimensional manifold, and let $x \in \mathcal{X}$ be a point. Then the tangent space $T_x \mathcal{X}$ consists of all $\gamma'(0)$, where $\gamma : (-\varepsilon, \varepsilon) \to \mathcal{X}$ is a smooth curve and $\gamma(0) = x$. The tangent bundle $T\mathcal{X}$ of $\mathcal{X}$ is the manifold whose underlying set is the disjoint union $\bigsqcup_{x \in \mathcal{X}} T_x \mathcal{X}$.

*Remark* 2.2. This definition does not tell us how to stitch $T_x \mathcal{X}$ and $T\mathcal{X}$ into manifolds. The details of that construction can be found in any introductory book on differential geometry. It suffices to note that $T_x \mathcal{X}$ is a vector space of dimension $d$, and $T\mathcal{X}$ is a manifold of dimension $2d$.

*Definition* 2.3. A Riemannian manifold is a pair $(\mathcal{X}, \langle \cdot, \cdot \rangle)$, where $\mathcal{X}$ is a manifold and $\langle \cdot, \cdot \rangle$ is a smoothly varying positive definite bilinear form on the tangent space $T_x \mathcal{X}$, for each $x \in \mathcal{X}$. We call $\langle \cdot, \cdot \rangle$ the (Riemannian) metric.

The Riemannian metric allows one to measure distances between two points on $\mathcal{X}$. We define the *length* of a curve $\gamma : [a, b] \to \mathcal{X}$ to be

$$L(\gamma) = \int_a^b \langle \gamma'(t), \gamma'(t) \rangle \, dt,$$

and the *distance* $\rho(x, y)$ to be

$$\rho(x, y) = \inf_{\substack{\gamma(0) = x \\ \gamma(1) = y}} L(\gamma).$$

A *geodesic* on a Riemannian manifold is a curve $\gamma : [a, b] \to \mathcal{X}$ that locally minimizes distance, in the sense that if $\widetilde{\gamma} : [a, b] \to \mathcal{X}$ is another path with $\widetilde{\gamma}(a) = \gamma(a)$ and $\widetilde{\gamma}(b) = \gamma(b)$ with $\widetilde{\gamma}(t)$ and $\gamma(t)$ sufficiently close together for each $t \in [a, b]$, then $L(\gamma) \leq L(\widetilde{\gamma})$.

*Example.* On $\mathbb{R}^d$ with the standard metric, geodesics are exactly the line segments, since the shortest path between two points is along a straight line.

In this article, we are primarily concerned with the case of $\mathcal{X}$ diffeomorphic to $\mathbb{R}^d$. However, it will be essential to think in terms of Riemannian manifolds, for our metric on $\mathcal{X}$ will vary from the standard metric. In §2.3, we will see how to choose a metric, the Jacobi metric, that is tailored to a non-uniform probability distribution $\pi$ on $\mathcal{X}$.

## 2.2 Hamiltonian Monte Carlo

In order to resolve some of the issues with the standard versions of MCMC related to slow mixing times, we draw inspiration from ideas in physics. We mimic the movement of a body under potential and kinetic energy changes to avoid diffusive behavior. The stationary probability will be linked to the potential energy. The reader is invited to read [15] for an elegant survey of the subject.

The setup is as follows: let $\mathcal{X}$ be a manifold, and let $\pi$ be a target distribution on $\mathcal{X}$. As with the Metropolis-Hastings algorithm, we start at some point $q_0 \in \mathcal{X}$. However, we use an analogue of the laws of physics to tell us where to go for future steps.

To simplify our exposition, we assume that $\mathcal{X} = \mathbb{R}^d$. This is not strictly necessary, but all distributions we consider will be on $\mathbb{R}^d$. In what follows, we let $(q_n, p_n)$ be the position and momentum after $n$ steps of the walk.

To run Hamiltonian Monte Carlo, we must first choose functions $V : \mathcal{X} \to \mathbb{R}$ and $K : T\mathcal{X} \to \mathbb{R}$, and we let $H(q, p) = V(q) + K(q, p)$. We start at a point $q_0 \in \mathcal{X}$. Now, supposing we have $q_n$, the position at step $n$, we sample $p_n$ from a $\mathcal{N}(0, I_d)$ distribution. We solve the differential equations

$$\frac{dq}{dt} = \frac{\partial H}{\partial p}, \qquad \frac{dp}{dt} = -\frac{\partial H}{\partial q} \tag{2.1}$$

with initial conditions $p(0) = p_n$ and $q(0) = q_n$, and we let $q_{n+1} = q(1)$.

In order to make the stationary distribution of the $q_n$'s be $\pi$, we choose $V$ and $K$ following Neal in [15]; we take

$$V(q) = -\log \pi(q) + C, \qquad K(p) = \frac{D}{2} \|p\|^2, \tag{2.2}$$

where $C$ and $D > 0$ are convenient constants. Note that $V$ only depends on $q$ and $K$ only depends on $p$. $V$ is larger when $\pi$ is smaller, and so trajectories are able to move more quickly starting from lower density regions than out of higher density regions.

## 2.3 Curvature

Not all probability distributions can be efficiently sampled. In particular, high-dimensional distributions such as the uniform distribution on the cube $[0, 1]^d$ are especially susceptible to sampling difficulties due to the curse of dimensionality, where in some cases it is necessary to take exponentially many (in the dimension of the space) sample points in order to obtain a satisfactory estimate. (See [13] for a discussion of the problems with integration on high-dimensional boxes and some ideas for tackling them when we have additional information about the function.)

However, numerical integration on high-dimensional spheres is not as difficult. The reason is that the sphere exhibits *concentration of measure*, so that the bulk of the surface area of the sphere lies in a small ribbon around the equator (see [14, §III.I.6]). As a result, we can obtain a good estimate of an integral on a high-dimensional sphere by taking many sample points around the equator, and only a few sample points far from the equator. Indeed, a polynomial number (in the dimension and the error bound) of points will suffice.

The difference between the cube and sphere, in this instance, is that the sphere has *positive curvature*, whereas the cube has zero curvature. Spaces of positive curvature are amenable to efficient numerical integration.

However, it is not just a *space* that can have positive (or otherwise) curvature. As we shall see, we can associate a notion of curvature to a Markov chain, an idea introduced by Ollivier [16] and Joulin [11] following work of Sturm [20, 21]. In this case as well, we will be able to perform numerical integration, using Hamiltonian Monte Carlo, in the case of stationary distributions of Markov chains with positive curvature. Furthermore, in §3, we will be able to provide error bounds for the integrals in question.

In order to make the geometry and the probability measure interdependent, we will deform our space to take the probability distribution into account, in a manner reminiscent of the inverse transform method mentioned in the introduction. Formally, this amounts to putting a suitable *Riemannian metric* on our state space $\mathcal{X}$. From now on, we shall assume that $\mathcal{X}$ is a *manifold*; in fact, it will generally suffice to let it be $\mathbb{R}^d$. Nonetheless, even in the case of $\mathbb{R}^d$, the extra Riemannian metric is important since it is not the standard Euclidean one.

Given a probability distribution $\pi$ on $\mathbb{R}^d$, we now define a metric on $\mathbb{R}^d$ that is tailored to $\pi$ and the Hamiltonian it induces (see §2.2). This construction is originally due to Jacobi, but our treatment follows Pin in [18].

*Definition* 2.4. Let $(\mathcal{X}, \langle \cdot, \cdot \rangle)$ be a Riemannian manifold, and let $\pi$ be a probability distribution on $\mathcal{X}$. Let $V$ be the potential energy function associated to $\pi$ by (2.2). For $h \in \mathbb{R}$, we define the Jacobi metric to be

$$g_h(\cdot, \cdot) = 2(h - V)\langle \cdot, \cdot \rangle.$$

*Remark* 2.5. $(\mathcal{X}, g_h)$ is not necessarily a Riemannian manifold, since $g_h$ will not be positive definite if $h - V$ is ever nonpositive. We could remedy this situation by restricting to the subset of $\mathcal{X}$ on which $h - V > 0$. However, this will not be problematical for us, as we will always select values of $h$ for which $h - V > 0$.

The reason for using the Jacobi metric is the following result of Jacobi, following Maupertuis:

**Theorem 2.6** (Jacobi-Maupertuis Principle, [10]). *Trajectories $q(t)$ of the Hamiltonian equations* 2.1 *with total energy $h$ are geodesics of $\mathcal{X}$ with the Jacobi metric $g_h$.*

The most convenient way for us to think about the Jacobi metric on $\mathcal{X}$ is as a distortion of space to suit the probability measure. In order to do this, we make regions of high density larger, and we make regions of low density smaller. However, the Jacobi metric does not completely override the old notion of distance and scale; the Jacobi metric provides a *compromise* between physical distance and density of the probability measure.

As we run Hamiltonian Monte Carlo as described in §2.2, $h$ changes at every step, as we let $h = V(q_n) + K(p_n)$. That is, we actually vary the metric structure as we run the chain, or, alternatively, move between different Riemannian manifolds. In practice, however, we prefer to think of the chain as running on a single manifold, with a changing structure.

We will not give all the relevant definitions of curvature, only a few facts that provide some useful intuition.

We will need the notion of *sectional curvature* in the plane spanned by $u$ and $v$. Let $\mathcal{X}$ be a $d$-dimensional Riemannian manifold, and $x, y \in \mathcal{X}$ two distinct points. Let $v \in T_x \mathcal{X}$, $v' \in T_y \mathcal{X}$ be two tangent vector at $x$ and $y$ that are related to each other by parallel transport along the geodesic in the direction of $u$. Let $\delta$ be the length of the geodesic between $x$ and $y$, and $\varepsilon$ the length of $v$ (or $v'$). Let $\rho$ be the length of the geodesic between the two endpoints starting at $x$ shooting in direction $\varepsilon v$, and $y$ in direction $\varepsilon v'$. Then the sectional curvature $\mathrm{Sec}_x(u, v)$ at point $x$ is given by

$$\rho = \delta \left( 1 - \frac{\varepsilon^2}{2} \mathrm{Sec}_x(u, v) + O(\varepsilon^3 + \varepsilon^2 \delta) \right) \text{ as } (\varepsilon, \delta) \to 0.$$

See Figure 3 in our long paper [9] for a pictorial representation.

We let Inf Sec denote the infimum of $\mathrm{Sec}_x(u, v)$, where $x$ runs over $\mathcal{X}$ and $u, v$ run over all pairs of linearly independent tangent vectors at $x$.

*Remark* 2.7. In practice, it may not be easy to compute Inf Sec precisely. As a result, we can approximate it by running a suitable Markov chain on the collection of pairs of linearly independent tangent vectors of $\mathcal{X}$; say we reach states $(x_1, u_1, v_1), (x_2, u_2, v_2), \ldots, (x_t, u_t, v_t)$. Then we can approximate Inf Sec by the *empirical* infimum of the sectional curvatures $\inf_{1 \le i \le t} \mathrm{Sec}_{x_i}(u_i, v_i)$. This approach has computational benefits, but also theoretical benefits: it allows us to ignore low sectional curvatures that are unlikely to arise in practice.

Note that Sec depends on the metric. There is a formula, due to Pin [18], connecting the sectional curvature of a Riemannian manifold equipped with some reference metric, with that of the Jacobi metric. We write down an expression for the sectional curvature in the special case where the reference metric on $\mathcal{X}$ is the standard Euclidean metric and $u$ and $v$ are orthonormal tangent vectors at a point $x \in \mathcal{X}$:

$$\mathrm{Sec}_x(u, v) = \frac{1}{8(h - V)^3} \Big( 2(h - V) \Big[ \langle (\mathrm{Hess}\, V) u, u \rangle + \langle (\mathrm{Hess}\, V) v, v \rangle \Big]$$
$$+ 3 \Big[ \| \operatorname{grad} V \|^2 \cos^2(\alpha) + \| \operatorname{grad} V \|^2 \cos^2(\beta) \Big] - \| \operatorname{grad} V \|^2 \Big). \quad (2.3)$$

Here, $\alpha$ is defined as the angle between $\operatorname{grad} V$ and $u$, and $\beta$ as the angle between $\operatorname{grad} V$ and $v$, in the standard Euclidean metric.

There is also a notion of curvature, known as *coarse Ricci curvature* for Markov chains [16]. (There is also a notion of Ricci curvature for Riemannian manifolds, but we do not use it in this article.) If $P$ is the transition kernel for a Markov chain on a metric space $(\mathcal{X}, \rho)$, let $P_x$ denote the transition probabilities starting from state $x$. We define the coarse Ricci curvature $\kappa(x, y)$ as the $W_1$ Wasserstein distance between two probability measures by

$$W_1(P_x, P_y) = (1 - \kappa(x, y)) \rho(x, y).$$

We write $\kappa$ for $\inf_{x, y \in \mathcal{X}} \kappa(x, y)$. We sometimes write $\kappa$ for an empirical infimum, as in Remark 2.7.

## 3 Concentration Inequality for General MCMC

We now state Joulin and Ollivier's [12] concentration inequalities for general MCMC. This will provide the link between geometry and MCMC that we will need for our concentration inequality for HMC.

*Definition* 3.1.
- The *Lipschitz norm* of a function $f : (\mathcal{X}, \rho) \to \mathbb{R}$ is

$$\|f\|_{\mathrm{Lip}} := \sup_{x, y \in \mathcal{X}} \frac{|f(x) - f(y)|}{\rho(x, y)}.$$

If $\|f\|_{\mathrm{Lip}} \le C$, we say that $f$ is $C$-Lipschitz.

- The *coarse diffusion constant* of a Markov chain on a metric space $(\mathcal{X}, \rho)$ with kernel $P$ at a state $q \in \mathcal{X}$ is the quantity

$$\sigma(q)^2 := \frac{1}{2} \iint_{\mathcal{X} \times \mathcal{X}} \rho(x, y)^2 \, P_q(dx) \, P_q(dy).$$

- The *local dimension* $n_q$ is

$$n_q := \inf_{\substack{f:\mathcal{X}\to\mathbb{R} \\ f \text{ 1-Lipschitz}}} \frac{\iint_{\mathcal{X}\times\mathcal{X}} \rho(x,y)^2 \, P_q(dx) \, P_q(dy)}{\iint_{\mathcal{X}\times\mathcal{X}} |f(x) - f(y)|^2 \, P_q(dx) \, P_q(dy)}.$$

- The *eccentricity* $E(q)$ at a point $q \in \mathcal{X}$ is defined to be

$$E(q) = \int_{\mathcal{X}} \rho(x,y) \, \pi(dy).$$

**Theorem 3.2** ([12]). *If $f : \mathcal{X} \to \mathbb{R}$ is a Lipschitz function, then*

$$|\mathbb{E}_q\widehat{I} - I| \leq \frac{(1-\kappa)^{T_0+1}}{\kappa T} E(q)\|f\|_{\mathrm{Lip}}.$$

**Theorem 3.3** ([12]). *Let*

$$V^2(\kappa, T) = \frac{1}{\kappa T}\left(1 + \frac{T_0}{T}\right) \sup_{q\in\mathcal{X}} \frac{\sigma(x)^2}{n_q\kappa}.$$

*Then, assuming that the diameters of the $P_q$'s are unbounded, we have*

$$\mathbb{P}_q(|\widehat{I} - \mathbb{E}_q\widehat{I}| \geq r\|f\|_{\mathrm{Lip}}) \leq 2e^{-r^2/(16V^2(\kappa,T))}.$$

Joulin and Ollivier [12] work with metric state spaces that have positive curvature. In contrast, in the next section, we work with Euclidean state spaces. We show that HMC transforms Euclidean state space into a state space with positive curvature. In HMC, curvature does not originate from the state space but from the measure $\pi$. The measure $\pi$ acts on the state space according to the rules of HMC; one can think of a distortion of the underlying state space, similar to the transform inverse sampling for one dimensional continuous distributions.

## 4 Concentration Inequality for HMC

In this section, we apply Theorem 3.3 for sampling from multivariate Gaussian distributions using HMC. For a book-length introduction to sampling from multivariate Gaussians, see [6]. We begin with a theoretical discussion, and then we present some simulation results. As we shall see, these distributions have positive curvature in high dimensions.

**Lemma 4.1.** *Let $C$ be a universal constant and $\pi$ be the $d$-dimensional multivariate Gaussian $\mathcal{N}(0,\Sigma)$, where $\Sigma$ is a $(d \times d)$ covariance matrix, all of whose eigenvalues lie in the range $[1/C, C]$. We denote by $\Lambda = \Sigma^{-1}$ the precision matrix. Let $q$ be distributed according to $\pi$, and $p$ according to a Gaussian $\mathcal{N}(0, I_d)$. Further, $h = V(q) + K(q,p)$ is the sum of the potential and the kinetic energy. The Euclidean state space $\mathcal{X}$ is equipped with the Jacobi metric $g_h$. Pick two orthonormal tangent vectors $u, v$ in the tangent space $T_q\mathcal{X}$ at point $q$. Then the sectional curvature $\mathrm{Sec}$ from expression (2.3) is a random variable bounded from below with probability*

$$\mathbb{P}(d^2 \, \mathrm{Sec} \geq K_1) \geq 1 - K_2 e^{-K_3\sqrt{d}}.$$

*$K_1$, $K_2$, and $K_3$ are positive constants that depend on $C$.*

We note that the terms in (2.3) involving cosines can be left out since they are always positive and small. The other three terms can be written as three quadratic forms in standard Gaussian random vectors. We then calculate tail inequalities for all these terms using Chernoff-type bounds. We also work out the constants $K_1$, $K_2$, and $K_3$ explicitly. For a detailed proof see our long paper [9].

There is a close connection between $\kappa$ and $\mathrm{Sec}$ of $\mathcal{X}$ equipped with the Jacobi metric: for Gaussians with assumptions as in Lemma 4.1, we have

$$\kappa \geq \frac{\mathrm{Sec}}{6d}.$$

as $d \to \infty$. We give the derivation in our long paper [9].

Now we can insert $\kappa$ into Theorem 3.3 and compute our concentration inequality for HMC. For details on how to calculate the coarse diffusion constant $\sigma(q)^2$, the local dimension $n_q$, and the eccentricity $E(q)$, see our long paper [9].

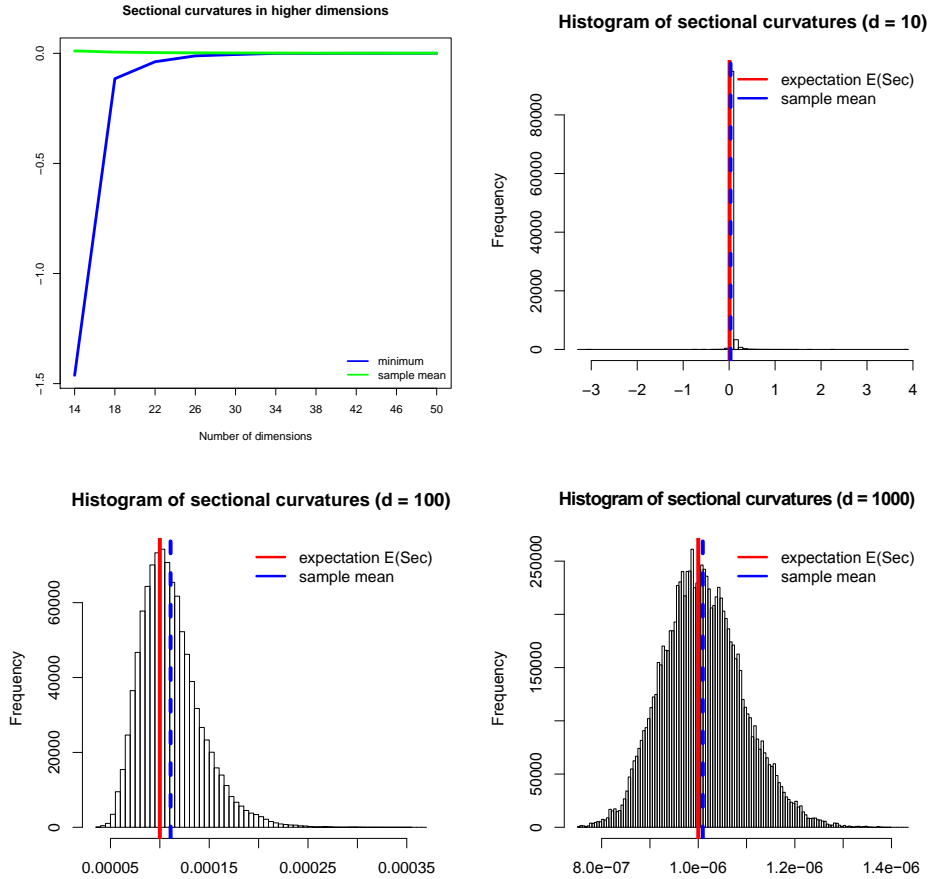

Figure 1: *Top left:* Minimum and sample average of sectional curvatures for 14- to 50-dimensional multivariate Gaussian $\pi$ with identity covariance. For each dimension we run a HMC random walk with $T = 10^4$ steps. *The other three plots:* HMC after $T = 10^4$ steps for multivariate Gaussian $\pi$ with identity covariance in $d = 10, 100, 1000$ dimensions. At each step we compute the sectional curvature for $d$ uniformly sampled orthonormal 2-frames in $\mathbb{R}^d$.

*Remark* 4.2. The coarse curvature $\kappa$ only depends on $\pi$. However, in practice we compute $\kappa$ empirically by running several steps of the chain as discussed in Remark 2.7, making $\kappa$ depend on $x$ and $T_0$. Thus, we typically assume $T_0$ to be known in advance in some other way.

*Example* (Distribution of sectional curvature). We run a HMC Markov chain to sample a multivariate Gaussian $\pi$. Figure 1 shows how the minimum and sample mean of sectional curvatures during the HMC random walk tend closer with dimensionality, and around dimension 30 we cannot distinguish them visually anymore. The minimum sectional curvatures are stable with small fluctuations. The actual sample distributions are shown in three separate plots (Figure 1) for 10, 100 and 1000 dimensions. These plots suggest that the sample distributions of sectional curvatures tend to a Gaussian distribution with smaller variances as dimensionality increases.

*Example* (Running time estimate). Now we give a concentration inequality simulation for sampling from a 100-dimensional multivariate Gaussian with with Gaussian decay between the absolute distance squared of the variable indices

$$\pi \sim \mathcal{N}(0, \exp(-|i - j|^2))$$

and the following parameters

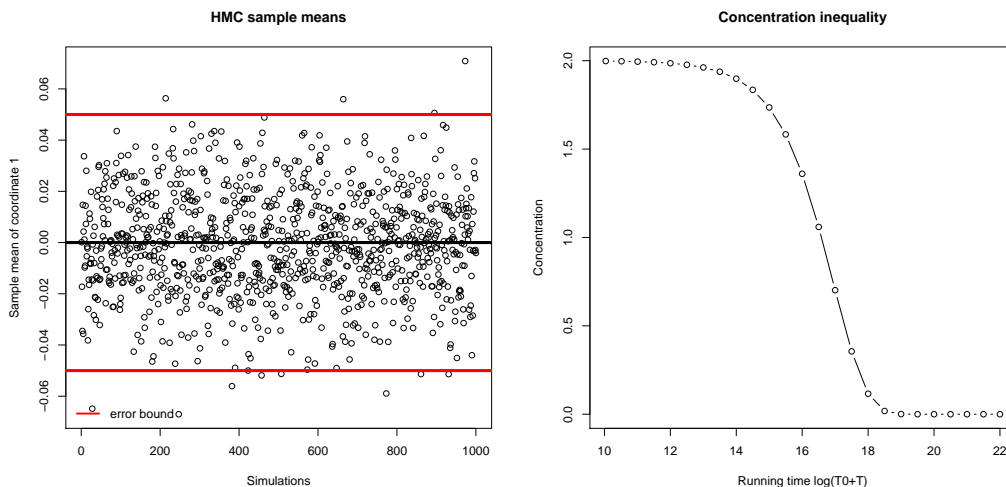

Figure 2: (Covariance structure with weak dependencies) Left: Sample means for 1000 simulations for the first coordinate of the 100 dimensional multivariate Gaussian. The red lines indicate the error bound $r$. Right: Concentration inequality with increasing burn-in and running time.

| Error bound | $r = 0.05$ | Starting point | $q_0 = 0$ |
|---|---|---|---|
| Markov chain kernel | $P \sim \mathcal{N}(0, I_{100})$ | Coarse Ricci curvature | $\kappa = 0.0024$ |
| Coarse diffusion constant | $\sigma^2(q) = 100$ | Local dimension | $n_q = 100$ |
| Lipschitz norm | $\|f\|_{\mathrm{Lip}} = 0.1$ | Eccentricity | $E(0) = 99.75$ |

For calculations of these parameters see our long paper [9]. In Figure 2 on the left, we show 1000 simulations of this HMC chain and for each simulations we plot the sample mean approximation to the integral. The red lines indicated the requested error bound at $r = 0.05$. From these simulation results, we would expect the right burn-in and running time to be around $T + T_0 = e^{10}$. In Figure 2 on the right, we see our theoretical concentration inequality as a function of burn-in and running time $T + T_0$ (in logarithmic scale). The probability of making an error above our defined error bound $r = 0.05$ is close to zero at burn-in time $T_0 = 0$ and running time $T = e^{19}$. The discrepancy between the predicted theoretical results and the actual simulations suggest there might be hope for improvements in future work.

## 5 Conclusion

Lemma 2.3 states a probabilistic lower bound. So in rare occasions, we will still observe curvatures below this bound or in very rare occasions even negative curvatures. Even if we had less conservative bounds on the number of simulations steps $T_0 + T$, we could still not completely exclude "bad" curvatures. For our approach to work, we need to make the explicit assumption that rare "bad" curvatures have no serious impact on bounds for $T_0 + T$. Intuitively, as HMC can take big steps around the state space towards the gradient of distribution $\pi$, it should be able to recover quickly from "bad" places. We are now working on quantifying this recovery behavior of HMC more carefully.

For a full mathematical development with proofs and more examples on the multivariate t distribution and in computational anatomy see our long paper [9].

### Acknowledgments

The authors would like to thank Sourav Chatterjee, Otis Chodosh, Persi Diaconis, Emanuel Milman, Veniamin Morgenshtern, Richard Montgomery, Yann Ollivier, Xavier Pennec, Mehrdad Shahshahani, and Aaron Smith for their insight and helpful discussions. This work was supported by a post-doctoral fellowship from the Swiss National Science Foundation and NIH grant R01-GM086884.

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
