[Reviews · NeurIPS 2014]

Submitted by Assigned_Reviewer_20

One of the most common reasons for using Markov chain Monte Carlo (MCMC) is to estimate the value of an otherwise intractable integral. Typically MCMC algorithms will give an exact answer as the number of iterations increases to infinity. However, this gives little assurance about the precision of the estimate in finite samples. This paper addresses this important issue from a theoretical point of view for the Hamiltonian Monte Carlo (HMC) algorithm, an algorithm which has been receiving a fair amount of recent attention. Moreover, the the theoretical approach taken, is interesting and relatively novel. Indeed, in their main result, the authors derive a concentration inequality for HMC that depends on coarse Ricci curvature and allows them to make statements about the precision of the estimated integral with high probability. These are impressive theoretical results.

Several other authors (Rosenthal, Baxendale, Meyn, Roberts, Tweedie, Latuszynki and Rudolf, to name a few) have attempted to provide similar types of results for standard MCMC algorithms such as the Metropolis random walk. While this previous work has been shown to be successful in a few applications, it is typically prohibitively difficult to apply in practically relevant MCMC settings. Moreover, the results are almost always too conservative to be useful in practice. As the presentation of the toy example in section 4 of the current paper shows, the methods herein are likely to suffer the same defects. Thus I have some concern about the usefulness of these new results outside of toy problems.

There are also some other issues of concern with the paper:

1. On almost every page there is notation that has not been fully (or at all) defined. This made the paper unduly difficult to follow.

2. There should be some discussion of the differences between the current paper and that of Joulin and Ollivier (Ann Prob, 2010), who also derive concentration inequalities for MCMC algorithms.

3. I did not understand the need for the emphasis on computational anatomy in section 1. This is never mentioned again in the paper. I kept waiting for an example to appear.

Summary: The paper presents some impressive theoretical results on the finite sample properties of HMC algorithms, but the results appear difficult to apply and overly conservative, which limits the significance of the work. Moreover, the paper could be more clearly written.

Submitted by Assigned_Reviewer_26

The authors state "The Jacobi metric connects seemingly different problems and enables us to transform a sampling
problem into a geometrical problem. It has been known since Jacobi [9] that Hamiltonian flows
correspond to geodesics on certain Riemannian manifolds. The Jacobi metric has been successfully
used in the study of phase transitions in physics; for a book-length account see [16]. But so far, to
the best of our knowledge, no connection has been made to probability and statistics."

In fact the explicit connection of the Maupertuis principle linking the local Jacobi metric and local Geodesic Flows to Hamiltonian flows on the cotangent bundle of a Riemannian manifold had already been previously highlighted and used in the development of methods for MCMC applications in probability and statistics in reference [7] - in particular see the rejoinder by the authors.

The results described in the submission are important and much needed to clarify understanding of sampling in high dimensional spaces. This is the frontier in many respects of statistical and machine learning methodology and application. The specific motivating example which the authors use in anatomical analysis is important and has similarities with many of the emerging application in ML such as Deep Learning etc. Therefore this analysis is important and timely.

Some previous analysis by mathematicians such as Andrew Stuart and co-workers have considered theoretical issues with highly significant practical import to sampling in high-dimensions by considering absolute continuity of measures (Gaussian) in Hilbert spaces - from which dimension independent proposal mechanisms emerged. To me it would seem that this theoretical analysis may well have similar impact further down the line - already so by exploiting the concentration bounds in quantifying burn-in required and running times - and so my view is highly favourable of this work.

Overall I found the NIPS version of the longer report to be somewhat condensed and concise but nevertheless the important points of the work emerge reasonably clearly after some little effort.
Summary: An important piece of theoretical analysis. This has strong practical implications. Well written and clear.

Submitted by Assigned_Reviewer_31

This paper seems to address the concentration in equalities for certain Markov chains. The authors focus on Hamiltonian MCMC. The presentation is somewhat unclear and seems to strew on various aspects of the setup, which to this reviewer, seem tangential to the main story. Whilst there may be some interesting underlying ideas I am unable to discern what is really going on and it seems the authors want to present a lot of complicated-sounding details without really getting to the point. Some basic issues:

1) Exactly what is the Hamiltonian MCMC algorithm under consideration? Does it involve exact or discretised solution of the differential equations?
2) The definition of kappa is buried in section 3 and I did not easily find it.
3) In the statement of Theorem 3.2, it sounds like kappa is somehow related to pi and f, but in the early definition of kappa, it is a functional of a Markov kernel. What is going on here?
4) Is the statement of Theorem 3.2, what does "running Hamiltonian Monte Carlo starting....and with transition kernel N(0,I_d)" mean? What is the algorithm in question here?
Summary: Presentation is overly complicated in some places, whilst in others key issues are glossed over or not presented with sufficient precision to be understandable.

Submitted by Assigned_Reviewer_34

This paper seem to have 3 main components:
1. The authors make quite a nice general setup for relating Ollivier's notion of curvature to some ideas in differential geometry.
2. The authors derive a straightforward application of Ollivier/Joulin's concentration result.
3. The authors do some careful computations getting very nice analytic answers for an important simple example.

The first seems quite important for bringing this to a general audience; the second is trivial but nice to write down; the third seems very important for showing that their approach really works (and proving that HMC `works well' for at least one example).
Since HMC is still not well understood even for simple examples, I welcome this theoretical contribution.

That said, there is one technical issues which vexes me; I am not even sure if the proposed methodology is sound because of this. The issue is that the authors estimate the minimum of curvatures from simulations. I am extremely unclear on how they are taking the "empirical infimum" of curvatures to get a lower bound on curvature and then apply bounds of Ollivier and Joulin. The bounds that they cite require the actual infimum, and it isn't at all clear to me that this empirical thing makes any sense. This sort of thing can certainly be resolved, but I don't know of any trivial way to do it, and the authors seem to gloss over the entire issue as far as I can see. They don't give even any intuition on why this must work.

Minor quibbles:
1. Remark 2.7 and Remark 3.3: In both of these remarks, the authors discuss calculating the curvature of the whole Markov chain by running the chain for a while and taking the sample infimum. This has a whole host of obvious problems. To mention a few: A. calculating an infimum via Monte Carlo is always dangerous in the absence of regularity assumptions, B. Calculating ANY measurement of mixing by running MCMC is even more dangerous, even with regularity assumptions, since the relevant chain may simply not have mixed. Of course, one can be lucky and everything can work out, but this seems like it careless to write down such a thing without any justification. Perhaps I am missing something here...

2. They say near the bottom of pp. 5 that $\kappa \approx \frac{ Inf Sec}{6}$. In Lemma 4.1, the sectional curvatures are random variables. How does one go from the distribution of a random variable to an infimum in this setting? How does this relate to the simulation at the end of the paper?

3. A few typos are scattered through out the paper; these must be corrected e.g., Equality 2.2: Sec(u,v) should be Sec_{x}(u,v)
Summary: An interesting theoretical contribution for HMC that shows promise, but in my opinion it might have a crucial flaw in the methodology. This must be explained/fixed before it can be published.
Author Feedback
Author rebuttal: We would like to thank the reviewers for their insightful feedback. Here is our response to their main concerns.

——————— (response to R20)
Usefulness outside of toy example

We agree with R20 that our concentration inequalities are very conservative. Nevertheless, it is a step towards understanding HMC. Moreover, our result on sectional curvature of HMC is important in its own right. We show in this paper that we can reduce the problem of analyzing HMC to analyzing a Markov chain on a high dimensional sphere with radius defined by the lower bound of sectional curvature. We are convinced that this will open HMC to further investigations using less conservative concentration inequalities.

Clear separation from Joulin and Ollivier’s work [11]

In the revised paper, we will untangle our contribution from Joulin and Ollivier’s [11] more explicitly. First, we will state Joulin and Ollivier’s concentration inequality for general MCMC (see response to R31, point 4), and second, we will state our new lemma 4.1 that proves a lower bound on sectional curvatures for HMC for the Gaussian example (see response to R34). We will also emphasize the different sources of curvature (see response to R31, point 3).
———————

——————— (response to R26)
Maupertuis principle

We thank R26 for the pointer to the discussion on the Maupertuis principle in the rejoinder of [7]. We will make sure to mention it in the revised paper.
———————

——————— (response to R31)
Clarity and precision

Here is our response to the four main issues raised by R31:

1) We study Hamiltonian Monte Carlo assuming that we can solve the Hamiltonian differential equations exactly. In contrast, [3] study optimal acceptance probability when discretizing the Hamiltonian equations using the leapfrog method.

2) The definition of kappa is at the end of section 2. In the revised paper, we will emphasize the importance of kappa and its infimum in section 3 with our new lemma 4.1 (see response to R34), which gives a rigorous lower bound on kappa.

3) We will make the connection between kappa, pi, f and HMC more precise. Everything hinges on the following key insight, which we will include in the revised paper:

Joulin and Ollivier [11] work with metric state spaces that have positive curvature. In contrast, we work with Euclidean state spaces. We show that HMC transforms Euclidean state space into a state space with positive curvature. In HMC, curvature does not originate from the state space but from the measure pi. The measure pi acts on the state space according to the rules of HMC; one can think of a distortion of the underlying state space, similar to the transform inverse sampling for one dimensional continuous distributions.

So yes, in HMC, kappa is linked to pi because pi acts on the Euclidean state space through the rules of HMC. This is a very important and very subtle point. I hope we could make it more clear now, and will make sure that this is emphasized in the revised paper.

4) We agree with R31 that theorem 3.2 was not clearly stated. We realized that a better way to state theorem 3.2 is by clearly separating it from our main result. Therefore, in the revised paper, we will replace theorem 3.2 with the general theorem from [11] (theorem 4, page 9). Our contribution is then postponed to section 4, where we show the link to HMC through our new lemma 4.1 (see response to R34).
———————

——————— (response to R34)
New lower bound on sectional curvatures for the Gaussian example

R34 raised concerns about our approach to compute infimums of sectional curvatures from simulations. To shed some light on this issue, we now have a new lemma that proves a lower bound on sectional curvatures for the Gaussian example. This will replace the weaker old lemma that proves mean and variance of sectional curvatures.

New Lemma 4.1:

Let C be a universal constant, let pi be the d-dimensional multivariate Gaussian N(0,Simga), where Sigma is a d x d covariance matrix, all of whose eigenvalues lie in the range [1/C, C]. Let Lambda = Sigma^-1 be the precision matrix. Let q be distributed according to pi, and p according to a Gaussian N(0,I_d). Further, let h = V(q) + K(q,p) be the sum of the potential and the kinetic energy. Let the Euclidean state space X be equipped with the Jacobi metric g_h. Pick two orthonormal tangent vectors u,v in the tangent space TqX at point q. Then the sectional curvature Sec from equation (2.2) is a random variable bounded from below with probability:
P(d^2 Sec \ge K_1) \ge 1 - K_2 exp(-K_3 sqrt(d)).
K_1, K_2, and K_3 are positive constants that depend on C.

Sketch of proof:

We note that the terms involving cosines can be left out since they are always positive and small. The other three terms can be written as three quadratic forms in standard Gaussian random vectors x,y,z (x and y are not independent of each other), giving us the inequality:
Sec \ge \frac{x^t Lambda x}{||p||^4 ||x||^2} + \frac{y^t Lambda y}{||p||^4 ||y||^2} - \frac{z^t Lambda z}{||p||^6}.
We then calculate tail inequalities for all these terms using Chernoff-type bounds. We also work out the constants K_1, K_2, and K_3 explicitly.

End of proof sketch.

This is a probabilistic lower bound. So in rare occasions, we will still observe curvatures below this bound or in very rare occasions even negative curvatures. Even if we had less conservative bounds on the number of simulations steps T0+T, we could still not completely exclude “bad” curvatures. For our approach to work, we need to make the explicit assumption that rare “bad” curvatures have no serious impact on bounds for T0+T. We will state this assumption more clearly in the revised paper. Intuitively, as HMC can take big steps around the state space towards the gradient of distribution pi, it should be able to recover quickly from “bad” places. We are currently working on quantifying this recovery behavior of HMC more carefully.
———————